# Is There a Difference in the Distribution of Mechanoreceptors among the Three Sections of the Anterior Talofibular Ligament?

**DOI:** 10.3390/medicina59091510

**Published:** 2023-08-22

**Authors:** Youngkoo Lee, Wonseok Park, Hyerim Lee, Youngsuk Choi, Sunghwan Kim, Euidong Yeo, Hongseop Lee, Kijin Jung, Byungryul Lee, Myoungjin Lee, Woojong Kim

**Affiliations:** 1Department of Orthopaedic Surgery, College of Medicine, Soonchunhyang University Bucheon Hospital, 170, Jomaru-ro, Wonmi-gu, Bucheon-si 14584, Republic of Korea; brain0808@hanmail.net (Y.L.); shk9528@naver.com (S.K.); 2Department of Orthopedic Surgery, Choonhae Hospital, 605 Jungang-daero, Busanjin-gu, Busan 47352, Republic of Korea; wsgodd@hanmail.net; 3Hyangseol Clinical Laboratory, Soonchunhyang University, Asan-si 31538, Republic of Korea; freehaerim3@naver.com (H.L.); imysuk@hanmail.net (Y.C.); 4Department of Orthopaedic Surgery, Veterans Health Service Medical Center, Seoul 05368, Republic of Korea; angel_doctor@naver.com; 5Nowon Eulji Medical Center, Department of Foot and Ankle Surgery, Eulji University, 68, Hangeulbiseok-ro, Nowon-gu, Seoul 01830, Republic of Korea; sup4036@naver.com; 6Department of Orthopaedic Surgery, Soonchunhyang University Hospital Cheonan, 31, Suncheonhyang 6-gil, Dongam-gu, Cheonan 31151, Republic of Korea; c89546@schmc.ac.kr (K.J.); 129027@schmc.ac.kr (B.L.); 7Department of Orthopaedic Surgery, Dong-A University Hospital, 26, Daesingongwon-ro, Seo-gu, Busan 49201, Republic of Korea; tynitus@dau.ac.kr

**Keywords:** ankle, anterior inferior talofibular ligament, distribution, mechanoreceptor

## Abstract

Background: We investigated whether the distribution of mechanoreceptors in three sections of the anterior talofibular ligament (ATFL) differed. Methods: The ATFL was obtained from 29 ankles of 21 fresh-frozen cadavers and divided into fibular attachment, mid-ligament, and talar attachment parts. Histologically, mechanoreceptors were classified as Ruffini (type I), Vater–Pacini (type II), Golgi–Mazzoni (type III), and free nerve ending corpuscles (type IV); the presence of these mechanoreceptors was compared among the three ATFL sections. Results: Type I mechanoreceptors were significantly more numerous than the other receptor types. Comparing the three sections of the ATFL, the number of type I mechanoreceptors differed significantly between the mid-ligament and fibular attachment (*p* = 0.006), while the number of type III mechanoreceptors differed significantly between the talar and fibular attachments (*p* = 0.005) and between the mid-ligament and talar attachment (*p* = 0.011). Conclusions: The four types of mechanoreceptors were distributed differently among the three sections of the ATFL. Type I mechanoreceptors were more numerous in all sections compared to the other receptors.

## 1. Introduction

Ankle sprains are the most common ankle injuries, with an incidence of 2.15 per 1000 people in the United States; nearly half of all ankle sprains occur during athletic activity [1]. Chronic ankle instability (CAI) can occur if ankle sprains are not treated properly with non-operative methods. Notably, approximately one-third of affected individuals have prolonged residual impairment, which affects functional activities (sports and daily living) [2,3]. CAI has been traditionally divided into two types: mechanical or functional instability. Mechanical instability can result from anatomical alterations, such as ligamentous laxity and arthrokinematic impairments. Functional instability can result from neuromuscular impairments, such as diminished proprioception and neuromuscular control [4]. Functional instability is when there are ongoing symptoms without any laxity in the joint that can be detected on examination [5]. However, it is now recognized that both mechanical and functional instability can exist together.

The ATFL originates from the anterior edge and tip of the lateral malleolus and runs anteriorly to insert at the neck of the talus. When the foot is in a neutral position, it is almost aligned with the axis of the foot, but when the foot bends downward, it follows a path that is parallel to the leg’s axis. Inversion sprains frequently damage this ligament, as they usually occur when the foot is in plantar flexion.

Articular denervation from a sprain can cause a proprioceptive deficit that leads to functional instability of the ankle. Instability and proprioception deficits occur in more than 20% of recurrent ankle sprains [6]. Proprioception refers to the capability of various mechanoreceptors to produce and send sensory signals to the brain pertaining to body position and posture; this plays a crucial role in balance control. Nociception refers to the perception of painful sensations [7,8]. Ankle proprioception, which may influence balance ability, can be changed by general and sport-specific training, sport-related injuries, and sport-induced fatigue [9,10]. Recent reviews have addressed the assessment of ankle proprioception in healthy individuals and subjects with musculoskeletal or neurological disorders [7,11].

To gain a better understanding of the function of the ankle collateral ligament, the mediating organs in the ligament have been examined microscopically. Freeman et al. hypothesized that mechanoreceptors, as specialized nerve endings, play a role in proprioception [4,12]. Functional ankle instability (FAI) results from damaged joint mechanoreceptors and proprioceptive deficits due to impaired ligament neuromuscular control, which occurs after ligament injury [4,12]. The treatment of FAI focuses on proprioceptive retraining, which determines the outcome [13,14]. There is no clear agreement on the morphology, classification, distribution, and mechanisms of the functional instability, which impedes the research on its treatment. In our previous research, we found that the distribution of mechanoreceptors varies even within a single ligament [15].

The most common site of damage during an ankle sprain is the fibula attachment, but damage can also occur in the mid-ligament or talar attachment or multiple sites. According to research by Kim et al. [16], when detecting the location of ATFL injury using arthroscopy and MRI, it was found that the fibular attachment was 40.0%, the midsubstance was 14.5%, the talar attachment was 27.3%, and multiple sites were 18.2%. Therefore, the authors wanted to investigate the distribution of mechanoreceptors in each area and assumed that if there were differences, it could affect treatment methods and outcomes. We hypothesized that there would be no significant difference in the distribution of mechanoreceptors among the three sections of the anterior talofibular ligament (ATFL). This study also investigated the types of mechanoreceptors present in these three sections. The identification of these mechanoreceptors would help provide a histological basis for the supposition that ligamentous differentiation underlies functional instability of the ankle.

## 2. Materials and Methods

This study was approved by the Institutional Review Board of Soonchunhyang University Bucheon Hospital (No. SCHBC2019-08-002-010).

A total of 29 ATFLs were harvested from 21 pairs of fresh-frozen human legs (Korean people, 12 male, 9 female cadavers; age range: 39–95 years; 15 right, 14 left). Non-intact ligaments (due to acute or chronic lesions, operation history) were excluded. The harvested ATFL was divided into three parts, each of which was defined as fibular attachment, mid-ligament, and talar attachment thirds. Tissue was processed and stained using a modified version of the gold chloride technique of Zimny et al. [17]. Each specimen was placed in a 3:1 mixture of fresh lemon juice and 98% formic acid and allowed to stand for 15 min; this process was repeated three times. Next, each specimen was placed in a 1% gold chloride solution (BioGnost, Zagreb, Croatia) and allowed to stand for 1.5 h. The specimens were placed in 2.5% formic acid overnight and then dehydrated in 30% sucrose for 2 days. Finally, each specimen was placed in a 3:2 mixture of 30% sucrose and optimal cutting temperature (OCT) compound for 2 h. The specimens were cut into 0.7-mm-wide × 0.5-mm-long pieces, frozen at −80 °C, cut into 30-μm-thick sections, and dried for 1 day. Finally, each specimen was dehydrated in a 60%, 70%, 90%, and 100% ethanol series (3 min each), and slides were prepared. A total of 72 slides were analyzed by obtaining 14 slides from each of the three parts of an ATFL.

All mechanoreceptors were counted under a microscope. The numbers of each receptor type in the three ATFL sections were counted twice each by two trained researchers (JYL and YSC). They were classified into four types by their morphology and function according to Freeman and Wyke: type I, slow Ruffini corpuscles; type II, dynamic Vater–Pacini corpuscles; type III, slow Golgi–Mazzoni corpuscles; and type IV, free nerve endings [18] (Table 1). To confirm the occurrence of mechanoreceptors, we evaluated the shape of the receptors in serial sections and non-artificial structures. We compared the numbers of the four receptor types among the three ATFL sections.

### Statistical Analysis

For statistical analysis, the Mann–Whitney and Kruskal–Wallis tests were used. The intraclass correlation coefficient (ICC) was used to calculate the intra- and interobserver reliability for repeated measurements. SPSS software (ver. 22.0; IBM Corp., Armonk, NY, USA) was used to perform all statistical analyses. A *p* < 0.05 was considered statistically significant. All reliability estimates are given with 95% confidence intervals (CIs).

## 3. Results

ATFLs were obtained from 29 ankles (Figure 1). We counted the four mechanoreceptor types in all tissues examined. Type I Ruffini was detected in clusters of three to six up to the thinly myelinated spindle corpuscles (Figure 2a). Type II Vater–Pacini formed thickly myelinated ovoid or cylindrical corpuscles in clusters of two to four (Figure 2b). Type III Golgi–Mazzoni was thinly myelinated, amorphous, and long and wide, with fusiform or spindle-shaped corpuscles (Figure 2c). Type IV free nerve endings were long, fine, non-myelinated fibers without a clearly defined shape (Figure 2d). Type I mechanoreceptors were significantly more numerous than the other receptor types (Figure 3). The mean numbers of type I–IV receptors were 1.56, 0.98, 1.27, and 1.16 in the fibular attachment section, 1.25, 0.68, 1.2, and 1.12 in the mid-ligament, and 1.44, 0.86, 0.75, and 1.29 in the talar attachment, respectively. The numbers of receptors were compared among the fibular attachment, mid-ligament, and talar attachment; the number of type I mechanoreceptors differed significantly between the mid-ligament and fibular attachment (*p* = 0.006), while the number of type III mechanoreceptors differed significantly between the talar and fibular attachments (*p* = 0.005) and between the mid-ligament and the talar attachment (*p* = 0.011) (Table 2 and Figure 3). Overall, there were significant differences in the numbers of type I and II mechanoreceptors in the fibular attachment section (*p* = 0.000), type I and III receptors (*p* = 0.034), and type I and IV receptors (*p* = 0.008). In the mid-ligament, there were significant differences in the numbers of type I and II receptors (*p* = 0.000), type II and III receptors (*p* = 0.001), and type II and IV receptors (*p* = 0.002). In the talar attachment, the numbers of type I and II mechanoreceptors (*p* = 0.001) and type I and III receptors (*p* = 0.002), type II and IV receptors (*p* = 0.022), and type III and IV receptors (*p* = 0.041) differed significantly (Table 3 and Figure 4). According to the intraclass correlation coefficients, paired comparisons were similar between observers. The counts of type I–IV mechanoreceptors had excellent intra- and interobserver reliability (0.873, 0.898, 0.863, and 0.903; and 0.876, 0.888, 0.825, and 0.846, respectively) (Table 4).

## 4. Discussion

The main finding of this study was that the various mechanoreceptors in the ATFL show differences in their distributions among the three sections thereof (the fibular attachment, mid-ligament, and talar attachment). Ruffini (type I), Vater–Pacini (type II), Golgi–Mazzoni (type III), and free nerve ending corpuscles (type IV) are all present in the ATFL [18]. Type I Ruffini corpuscles are slowly adapting receptors with a low threshold that respond to mechanical stress; they are often categorized as static or dynamic mechanoreceptors, transmitting information regarding the static position, changes in intra-articular pressure, and the direction, amplitude, and velocity of joint movement. Type II Vater–Pacini corpuscles are dynamic, rapidly adapting receptors with a low threshold; they become active only at the onset or cessation of joint motion. Type III Golgi–Mazzoni corpuscles are also dynamic receptors but have a high threshold and slow adaptation. They are completely inactive in immobile joints, becoming active only at the extremes of their ranges of motion and when under considerable stress. Type IV free nerve endings have a high threshold and are classified as non-adapting nociceptive receptors [18,19].

Michelson and Hutchins [20] investigated the types of mechanoreceptors in five cadaver ankle ligaments. They found that type II and III mechanoreceptors were the most numerous in the ATFL, while some type I receptors were found in all ankle ligaments.

Moraes et al. [8] studied mechanoreceptors in the anterior talofibular, posterior talofibular (PTFL), and calcaneofibular (CFL) ligaments, which are lateral ankle ligaments, in 24 cadaver ankles. The presence of mechanoreceptors in the lateral ankle ligaments has clinical significance and is relevant for proprioceptive function, as suggested by the innervation. They found that type II mechanoreceptors were more numerous than types I and III. However, there was no significant difference in mechanoreceptor density among the three different ligaments analyzed. In addition, they suggested that the presence of mechanoreceptors could have clinical implications and relevance for proprioceptive function, and future electrophysiological studies will be needed to define their role in the proprioceptive and nociceptive systems of the ankle. According to Moraes et al. [8], there were no significant differences in the total density of mechanoreceptors among the ATFL, CFL, and PTFL. However, Michelson and Hutchins found a significant difference between the CFL and PTFL (*p* < 0.005) but not between the ATFL and the CFL or PTFL [20].

Wu et al. [21] harvested the ATFL, PTFL, and CFL ligaments from six fresh frozen cadavers and identified four typical nerve ending types; type II receptors predominated and did not significantly differ in density among the distal, middle, and proximal ligament sections. They found that this indicates that the main function of ankle collateral ligaments is to sense joint speeds in motions. These findings provide morphological evidence for the presence of proprioceptive mechanoreceptors in ligaments, which could aid in the treatment of lateral ankle sprains and chronic ankle instability. They concluded that the morphology and distribution of the mechanoreceptors differed from what was reported in the literature, suggesting that their proprioceptive functions were more complex in humans. Additionally, they also stated that future research into the function of mechanoreceptors could lead to the development of new strategies for treating ligament injuries in ankle joints.

Yeo et al. [15] found that the four mechanoreceptor types were distributed in the Bassett’s ligament, the ATFL, and the ankle joint synovium in a cadaver study. There was no significant difference in the distribution of the mechanoreceptors among these three structures, except that the type I receptor density was significantly higher in the Bassett’s ligament. Through their research, they identified many mechanoreceptors that could play a role in ankle stability in the Bassett’s ligament and argued that the Bassett’s ligament should be carefully resected and not removed during arthroscopic ATFL repair. In our study, there were significantly more type I mechanoreceptors in all sections of the ATFL compared to the other types. This differs from the above studies reporting more type II receptors in the ATFL [8,20,21]. Unlike Wu et al. [21], we also found that the distribution of mechanoreceptors differed among the three ATFL sections. The morphology and distribution of these mechanoreceptors differed from what has been reported in the literature, suggesting that their proprioceptive functions in humans may be more complex. In addition, the difference among the studies might be due to differences in the number of samples investigated, examiner subjectivity, and subject ethnicity.

Proprioception is the process by which the body’s nervous system receives and interprets sensory information from the environment to produce a coordinated motor response [22]. When the knee or ankle is injured, the ligaments and their sensory receptors are also damaged, which affects the ability to sense the position and movement of the joint [12,23]. To recover from such injuries, the rehabilitation program should not only focus on the structural problem but also on restoring the lost sensory feedback [24]. There are specific exercises for ankle rehabilitation that aim to improve proprioception after ankle sprains, and these have been proven to be more effective than rehabilitation without proprioceptive training [25]. Proprioceptive training for the ankle joint can include exercises such as single-leg balancing with eyes closed, wobble board or ankle disk balancing, and single-leg balancing while performing a task such as catching or throwing a ball. These exercises can improve the sensorimotor system’s ability to adapt to changes in the environment and help prevent injury. Proprioceptive training can reduce a patient’s risk of sustaining a first-time or recurrent ankle sprain. Proprioceptive training is an efficient and cost-effective intervention that can help patients who have previously experienced an ankle sprain during physical activity. This type of training can also reduce the risk of future complications [26].The lateral ligaments of the ankle include the anterior talofibular ligament (ATFL), the calcaneofibular ligament (CFL), and the posterior talofibular ligament (PTFL). The ATFL is contiguous with the joint capsule and has a variable number of bands. In a study of human lateral ankle ligaments using cadavers, the authors discovered that half (7 out of 14) of the samples had a single-band ATFL, while the rest had two bands [27]. A single-banded ATFL’s origin is located on the fibula’s anterior margin approximately 13.8 mm from the lateral malleolus’s inferior tip and inserts just anterior to the lateral articular surface of the talus [27]. The CFL is a ligament that begins about 5.3 mm from the anterior tip of the lateral malleolus and attaches to the calcaneus. It passes under the peroneal tendons and crosses both the subtalar and talocrural joints. The CFL courses posteroinferiorly to insert onto the lateral aspect of the calcaneal body [28]. The PTFL is a ligament that starts from the medial side of the lateral malleolus, about 4.8 mm above its inferior tip [27].

The ATFL, CFL, and PTFL are ligaments that provide stability to the lateral aspect of the ankle joint. The ATFL is tight when the foot is in plantarflexion and prevents anterior talar displacement and excessive plantarflexion [29]. The ATFL is at risk of rupture following an inversion injury to a plantarflexed foot. It is also the most fragile of the three ligaments.

Ankle sprains are very common injuries among the most common injuries in the physically active population. A prior history of ankle sprain is the most prevalent factor that predisposes one to experience another ankle sprain [30]. The sensation of the ankle collapsing after the first ankle sprain and repeated episodes of instability that cause many more ankle sprains are called CAI. CAI has many different contributing factors, including deficits in postural control [31]. Ankle sprains can cause damage to the muscles and tendons and interfere with the inherent mechanoreceptors. This can reduce the quality of the proprioceptive information that is essential for balance control. Ankle proprioception may be impaired without rehabilitation following an ankle injury. This may result in long-term deficits in postural and balance control [32].

Ankle sprains are usually categorized into three grades based on the severity of the injury in clinical practice: grade I (mild), grade II (moderate), and grade III (severe) injuries. In grade I injuries, the ligaments are stretched but not torn, there is little swelling or tenderness, and there is no functional loss or joint instability. In grade II injuries, the ligaments are partially torn, and there is moderate pain, swelling, and tenderness over the injured structures. There is some loss of joint motion and mild to moderate joint instability. In grade III injuries, the ligaments are completely torn, and there is marked pain, swelling, hemorrhage, and tenderness. There is a loss of function and marked abnormal joint motion and instability.

Nonoperative management is the preferred option for grade I and II injuries, as most clinicians agree that it results in fast recovery and a very good or excellent outcome. The treatment program, called “functional treatment”, involves applying the RICE principle (rest, ice, compression, and elevation) right after the injury, a brief period of immobilization and protection with an elastic or inelastic tape or bandage, and early exercises to restore range of motion, followed by early weight bearing and neuromuscular training for the ankle. Proprioceptive training aims to improve the balance and neuromuscular control of the ankle. The balance and neuromuscular control of the ankle can be improved by starting proprioceptive training with a tilt board as soon as possible. Prospective, randomized studies have demonstrated the effectiveness of tilt board training [4,33,34,35], and additional muscle exercises, especially peroneus muscle strengthening, are recommended. It has been reported that the healing outcomes of lateral ligament injuries are related to functional instability, not mechanical instability [36].

Ligament damage due to ankle sprains occurs most frequently at the fibula attachment, but it can also occur at the mid-ligament or talar attachment. The authors suggest that the rehabilitation methods may need to vary or be refined depending on the injured site because the distribution of mechanoreceptors is different in each area. To achieve this, additional research may be needed to develop rehabilitation programs tailored to the specific site of injury.

Our study had some limitations. First, errors may occur when manually counting mechanoreceptors. The free nerve endings were difficult to distinguish from other nerve bundles given their small size [37]. Michelson and Hutchins [20] found no type Ⅳ free nerve endings in a mechanoreceptor study of five cadaver ankles. The larger Ruffini corpuscles have also been reported to resemble Golgi–Mazzoni corpuscles [38]. Moraes et al. [8] demonstrated that free nerve endings were frequently of the Ruffini type. Therefore, in future studies, it is considered necessary to aim at automatically and accurately counting mechanoreceptors, either technically or by introducing a program, rather than manually. This will require further study with histology experts. Second, the study has limited clinical relevance; the frequencies and complications of injuries to the three sections of the ATFL should be studied. Third, the sample size was small, and a larger multicenter study is needed. Our study cannot represent the distribution of ATFL mechanoreceptors in the global population. In addition, the distribution of mechanoreceptors can be different in gender, age, and direction of the ankle, which requires more sample studies in the future.

## 5. Conclusions

There were differences in the distribution of mechanoreceptors among the three sections of the ATFL. Type I mechanoreceptors were more common in all three sections compared to the other receptors. We suggest that rehabilitation of the ATFL should be based on the injured part and types of mechanoreceptors affected. The authors believe that more advanced research on mechanoreceptors responsible for proprioception is needed. In addition, research on refined rehabilitation programs can also contribute to the treatment of ATFL injuries in the future.

## Figures and Tables

**Figure 1 medicina-59-01510-f001:**
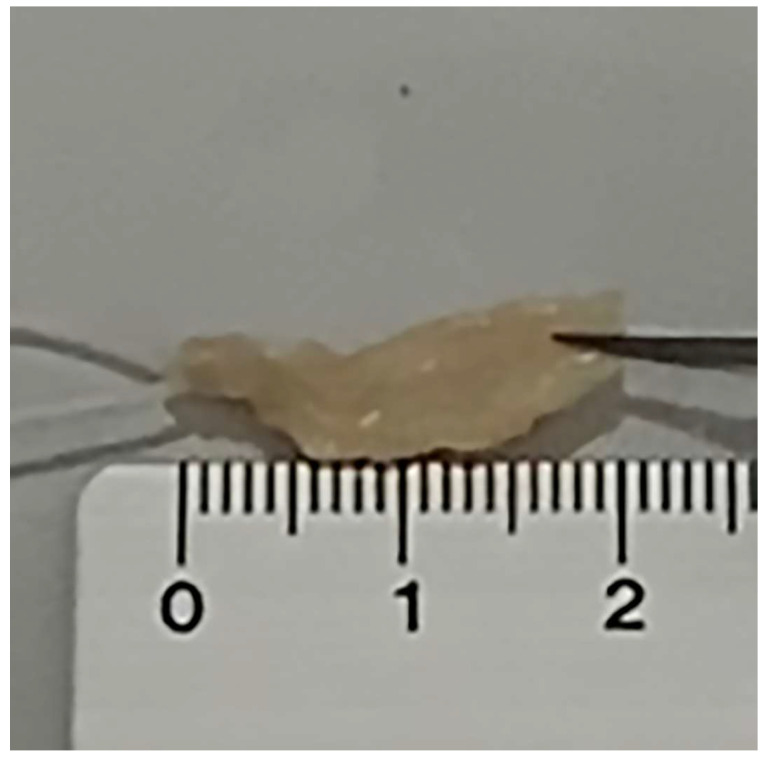
Resected ATFL ligament specimen.

**Figure 2 medicina-59-01510-f002:**
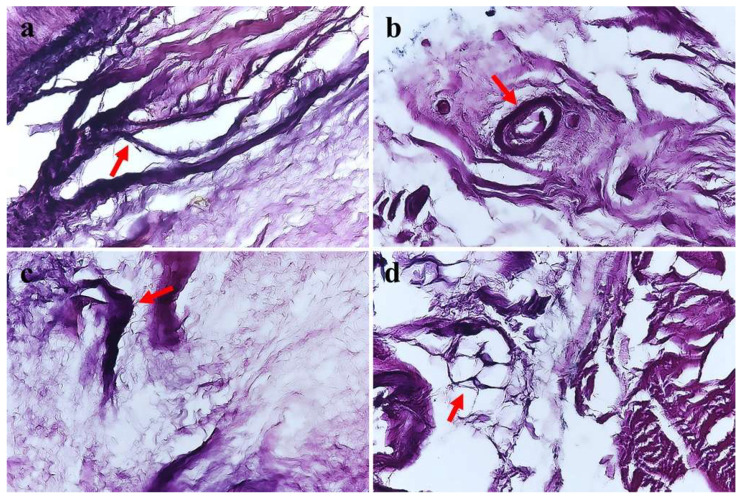
(**a**) Photomicrograph of a Ruffini corpuscle, detected in clusters of three to six (×200). (**b**) Photomicrograph of a Vater–Pacini corpuscle, detected in clusters of two to four with an ovoid or cylindrical shape (×200). (**c**) Photomicrograph of a Golgi–Mazzoni corpuscle, which is thinly myelinated, amorphous, long and wide, and fusiform or spindle-shaped (×200). (**d**) Photomicrograph of free nerve ending corpuscles, which are long, fine, and non-myelinated without a clearly defined shape (×200). Red arrows in each picture indicate each mechanoreceptor.

**Figure 3 medicina-59-01510-f003:**
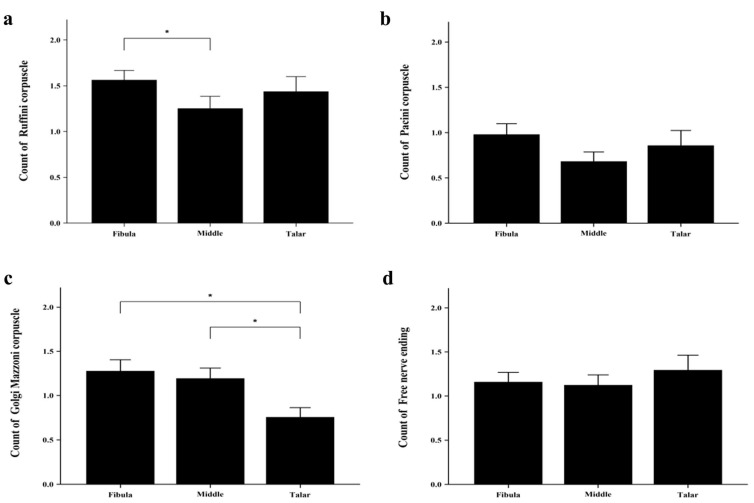
Mechanoreceptor counts. (**a**) Ruffini; (**b**) Vater–Pacini; (**c**) Golgi–Mazzoni; and (**d**) Free nerve ending corpuscles. The four types of mechanoreceptors in the different ligament sections were counted. * significant difference in the number of mechanoreceptors between ligament sections. (*p* < 0.05).

**Figure 4 medicina-59-01510-f004:**
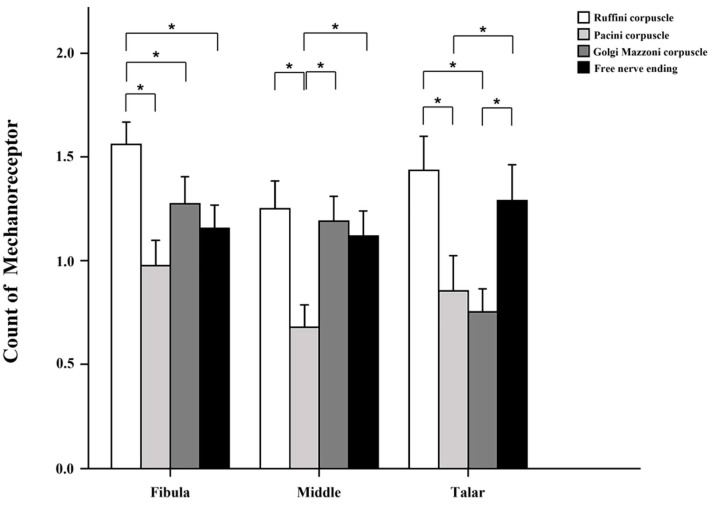
Mechanoreceptor counts according to ligament sections. The four types of mechanoreceptor were counted in the three ligament sections. * significant difference in the number of mechanoreceptors between ligament sections (*p* < 0.05).

**Table 1 medicina-59-01510-t001:** Freeman and Wyke mechanoreceptor classification.

Type	Name	Morphology	Characteristics
I	Ruffini	Round shaped thinly myelinated globular corpuscles in clusters of 3 to 6	Low threshold, slow adapting, static and dynamic
II	Pacini	Column or cone-shaped thickly myelinated corpuscles in clusters of 2 to 4	Low threshold, rapidly adapting, dynamic
III	Golgi	Spindle-shaped thinly myelinated, and connected by thick nerve fibers	Low threshold, slow adapting, dynamic
IV	Free nerve ending	Non-myelinated, irregular	Transmit nociceptive sensation

**Table 2 medicina-59-01510-t002:** Relative abundance of the mechanoreceptor types.

Receptor	Section	Mean ± SD	*p*-Value Compared to M	*p*-Value Compared to T
Ruffini	F	1.56 ± 1.0	0.006	
	M	1.25 ± 1.23		
	T	1.44 ± 1.37		
Pacini	F	0.98 ± 1.12		
	M	0.68 ± 0.98		
	T	0.86 ± 1.41		
Golgi	F	1.27 ± 1.2		0.005
	M	1.2 ± 1.1		0.011
	T	0.75 ± 0.91		
Free nerve ending	F	1.16 ± 1.04		
	M	1.12 ± 1.10		
	T	1.29 ± 1.44		

Abbreviations: F, fibular attachment; M, mid-ligament; SD, standard deviation; T, talar attachment. Results of comparisons of mechanoreceptor types using the Kruskal–Wallis and Mann–Whitney U tests. Significant differences between groups are shown (*p* < 0.05).

**Table 3 medicina-59-01510-t003:** Relative abundance of the mechanoreceptors according to ATFL sections.

Sections	Receptor	Mean ± SD	*p* Value Compared to V	*p* Value Compared to G	*p* Value Compared to F
Fibular attachment	R	1.56 ± 1.0	0.000	0.034	0.008
	V	0.98 ± 1.12			
	G	1.27 ± 1.2			
	F	1.16 ± 1.04			
Mid-ligament	R	1.25 ± 1.23	0.000		
	V	0.68 ± 0.98		0.001	0.002
	G	1.2 ± 1.1			
	F	1.12 ± 1.10			
Talar attachment	R	1.44 ± 1.37	0.001	0.002	
	V	0.86 ± 1.41			0.022
	G	0.75 ± 0.91			0.041
	F	1.29 ± 1.44			

Abbreviations: ATFL, anterior talofibular ligament; F, free nerve ending corpuscles; G, Golgi–Mazzoni; R, Ruffini; SD, standard deviation; V, Vater–Pacini. ATFL sections were compared in terms of mechanoreceptor abundance using the Kruskal–Wallis and Mann–Whitney U tests. Significant differences between groups are shown (*p* < 0.05).

**Table 4 medicina-59-01510-t004:** Intra- and interobserver reliability of the mechanoreceptor counts.

	Intraobserver Reliability		Interobserver Reliability	
	ICC	95% ± CI	ICC	95% ± CI
Ruffini	0.873	0.823–0.913	0.876	0.819–0.917
Vater–Pacini	0.898	0.858–0.930	0.888	0.838–0.925
Golgi–Mazzoni	0.863	0.810–0.906	0.825	0.783–0.911
Free nerve ending	0.903	0.866–0.934	0.846	0.776–0.897

Abbreviations: CI, confidence interval; ICC, intraclass correlation coefficient.

## Data Availability

The datasets used and/or analyzed during the current study are available from the corresponding author upon reasonable request.

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
