# Peer review of "Is There a Difference in the Distribution of Mechanoreceptors among the Three Sections of the Anterior Talofibular Ligament?"

_medicina, 2023, doi:10.3390/medicina59091510_

Round 1
Reviewer 1 Report
This study is that they are well-structured and well-written. They communicate their findings clearly and concisely to the reader. This study has an interesting research problem about ankle instability involving the distribution of receptors that may have potential therapeutic applications. The study objectives were consistent with the hypothesis. The assumptions in this study are consistent with previous studies. The results of the study are clearly demonstrated and measurable.
However, for the scope of research in part of materials and methods, there is no specified population or ethnicity which really should be mentioned because the researcher is mentioned in the discussion section. From the description of ATFL was divided into fibular attachment, mid-ligament, and talar attachment is not clearly visible whether divides three parts equally, should the diagram of these dividing three parts be shown. A sample size of 29 ankles might not be sufficiently representative of the population, and it doesn't explain why this number was chosen, such as that it has been determined to be an appropriate number from the calculation. Manual counting of receptors may not be sufficient to obtain accurate data because the counter may have errors, and two counts may not be enough. The Results section, from page 3 lines 95 to 100 describes the results that are inconsistent with the data in Table 2 Moreover, Table 2: Relative abundance of the mechanoreceptor types and Table 3. Relative abundance of the mechanoreceptors according to ATFL sections that show the numbers, and the data appear unusual because they are all the same set of numbers. Therefore, there is a question in these 2 tables, how are they different? In the discussion section: errors may occur when manually counting mechanoreceptors. There should be further discussion on ways to fix it, such as how technology or programs might be introduced in the future to reduce the error, and that The free nerve endings were difficult to distinguish from other nerves, there should be further discussion on ways to fix it, such as further to find ways to differentiate the receptor by consultation with histology experts. In applying it in the future, it is not clear or sufficiently specific enough. Additionally, there might be some interesting issues with the receptor distribution, such as that the distribution may be varied by sex, age, and the left and right ankles which weren't mentioned in the discussion part.
This study is that they are well-structured and well-written. They communicate their findings clearly and concisely to the reader. This study has an interesting research problem about ankle instability involving the distribution of receptors that may have potential therapeutic applications. The study objectives were consistent with the hypothesis. The assumptions in this study are consistent with previous studies. The results of the study are clearly demonstrated and measurable.
However, for the scope of research in part of materials and methods, there is no specified population or ethnicity which really should be mentioned because the researcher is mentioned in the discussion section. From the description of ATFL was divided into fibular attachment, mid-ligament, and talar attachment is not clearly visible whether divides three parts equally, should the diagram of these dividing three parts be shown. A sample size of 29 ankles might not be sufficiently representative of the population, and it doesn't explain why this number was chosen, such as that it has been determined to be an appropriate number from the calculation. Manual counting of receptors may not be sufficient to obtain accurate data because the counter may have errors, and two counts may not be enough. The Results section, from page 3 lines 95 to 100 describes the results that are inconsistent with the data in Table 2 Moreover, Table 2: Relative abundance of the mechanoreceptor types and Table 3. Relative abundance of the mechanoreceptors according to ATFL sections that show the numbers, and the data appear unusual because they are all the same set of numbers. Therefore, there is a question in these 2 tables, how are they different? In the discussion section: errors may occur when manually counting mechanoreceptors. There should be further discussion on ways to fix it, such as how technology or programs might be introduced in the future to reduce the error, and that The free nerve endings were difficult to distinguish from other nerves, there should be further discussion on ways to fix it, such as further to find ways to differentiate the receptor by consultation with histology experts. In applying it in the future, it is not clear or sufficiently specific enough. Additionally, there might be some interesting issues with the receptor distribution, such as that the distribution may be varied by sex, age, and the left and right ankles which weren't mentioned in the discussion part.
Author Response
Response to Reviewer 1 Comments
First of all, thank you for the good review.
Comments and Suggestions for Authors
This study is that they are well-structured and well-written. They communicate their findings clearly and concisely to the reader. This study has an interesting research problem about ankle instability involving the distribution of receptors that may have potential therapeutic applications. The study objectives were consistent with the hypothesis. The assumptions in this study are consistent with previous studies. The results of the study are clearly demonstrated and measurable
Response 1:
the scope of research in part of materials and methods, there is no specified population or ethnicity which really should be mentioned because the researcher is mentioned in the discussion section.
: We agree with your point.
We mentioned ethnicity in line 56 of the material and method, and in the limitation part of the discussion, it can vary depending on ethnicity and race. Thank you.
From the description of ATFL was divided into fibular attachment, mid-ligament, and talar attachment is not clearly visible whether divides three parts equally, should the diagram of these dividing three parts be shown.
: As the reviewer pointed out, it is difficult to distinguish precisely. We acknowledge that this can be a limitation in ligamant research. Therefore, we divided the entire length of ATFL equally into three parts and defined it as fibular attachment, mid-ligament, and talar attachment. We appreciate the reviewer's understanding of the limitations of the cadaver experiment. This is mentioned in lines 58-59.
Thank you.
A sample size of 29 ankles might not be sufficiently representative of the population, and it doesn't explain why this number was chosen, such as that it has been determined to be an appropriate number from the calculation
: We agree with your point. The sample size of 29 is not representative of the entire population. Referring to other studies, the Cadeva study has limitations in sample size, but we think it is a similar or larger number of samples compared to other studies. The limitation of sample size was mentioned in the limitation part. Thank you.
Therefore, there is a question in these 2 tables, how are they different?
: We agree with your points. Sorry for the inconvenience There was a mistake in the process of uploading Table 2. We re-uploaded the correct Table 2. Thank you for your understanding with the broad generosity of the reviewer.
In the discussion section: errors may occur when manually counting mechanoreceptors. There should be further discussion on ways to fix it, such as how technology or programs might be introduced in the future to reduce the error, and that The free nerve endings were difficult to distinguish from other nerves, there should be further discussion on ways to fix it, such as further to find ways to differentiate the receptor by consultation with histology experts.
: We agree with your points. As you mentioned, we added it to the limitation part.
Additionally, there might be some interesting issues with the receptor distribution, such as that the distribution may be varied by sex, age, and the left and right ankles which weren't mentioned in the discussion part.
: : We agree with your points. As you mentioned, we added points to the third limitation part. More research with a larger sample size is likely to be needed in the future.
Thank you again for your thoughtful and great review.
The authors would like to thank the reviewer once again for the excellent advice.

Reviewer 2 Report
Thank you for the opportunity of reviewing the manuscript titled "Is there a Difference in the Distribution of Mechanoreceptors 2 among the Three Sections of the Anterior Talofibular Liga-3 ment?". I have carefully read the paper and recommend some modifications before it could be considered for publication. Please find my comments below:
ABSTRACT
Ok
INTRODUCTION
Although the authors make an adequate "art status" regarding proprioception, mechanoreceptors, and how they may be injured because of ankle sprain and, therefore, affect general balance, I think that the reason why It Is Important to know the distribution of mechanoreceptors among the 3 sections of the anterior talofibular ligament has not sufficient justification. Authors should delve deeper into this topic.
This section Is too short and needs more argument, antecedents, etc.
MATERIALS AND METHODS
Line 56, How can authors ensure that the 29 ligaments Included were healthy ligaments, without previous ankle sprains. Please clarify.
The authors sustain that the specimens were cut Into 0.7-mm-wide x 0.5-mm-long pieces, and then cut Into 30-μm-thick sections. How many pieces were obtained from each specimen? How many of these pieces were analyzed to count the mechanoreceptors? Please clarify.
Line 77, the sentence "The numbers of each receptor type in the three ATFL sections were counted twice 77 each by two trained researchers (JYL, YSC)" should not constitute a part of the statistical analysis paragraph, but a part of the procedure previously explained.
RESULTS
Arrows are difficult to see In figure 2. Please make them bigger and with a different color.
P-values should be limited to 3 decimals.
DISCUSSION
The results are poorly discussed. Why Is Important to know how many mechanoreceptors there Is In each section of the ligament? Which are the clinical Implications of these findings?
I think this manuscript Is more suitable for an Anatomy journal.
CONCLUSIONS
Authors suggest that rehabilitation of the ATFL should be based on the injured part and types of mechanoreceptors affected. How can be rehabilitated an Independent part of the ligament or some types of mechanoreceptors only should be part of the discussion section.
REFERENCES
OK.
TABLES
P-values should be limited to 3 decimals.
Author Response
Response to Reviewer 2 Comments
First of all, thank you for the good review.
Comments and Suggestions for Authors
Thank you for the opportunity of reviewing the manuscript titled "Is there a Difference in the Distribution of Mechanoreceptors 2 among the Three Sections of the Anterior Talofibular Liga-3 ment?". I have carefully read the paper and recommend some modifications before it could be considered for publication. Please find my comments below:
Response 1:
INTRODUCTION
Although the authors make an adequate "art status" regarding proprioception, mechanoreceptors, and how they may be injured because of ankle sprain and, therefore, affect general balance, I think that the reason why It Is Important to know the distribution of mechanoreceptors among the 3 sections of the anterior talofibular ligament has not sufficient justification. Authors should delve deeper into this topic.
This section Is too short and needs more argument, antecedents, etc.
: We agree with your point. The introduction has been further supplemented. thank you
MATERIALS AND METHODS
Line 56, How can authors ensure that the 29 ligaments Included were healthy ligaments, without previous ankle sprains. Please clarify.
: Thanks for the pointed out points. First, the surgical history was confirmed. And 42 samples were taken, but 29 samples were obtained, excluding cases where there was too thinning, no deformation, or deformation. In fact, as the reviewer pointed out, In fact, as the reviewer pointed out, it cannot be absolutely declared a healthy ligament with no ankle sprain at all. However, the experiment was conducted using what appeared to be a healthy ligament as much as possible. We appreciate the reviewer's understanding of the limitations of the cadaver experiment.
: As pointed out by reviewer, line 58 has also been modified by adding "operation history".
The authors sustain that the specimens were cut Into 0.7-mm-wide x 0.5-mm-long pieces, and then cut Into 30-μm-thick sections. How many pieces were obtained from each specimen? How many of these pieces were analyzed to count the mechanoreceptors? Please clarify.
: A total of 72 slides were analyzed by obtaining 14 slides from each of the three parts of an ATFL. Thank you As pointed out, we added this sentence to the ‘material and method’ section.
Line 77, the sentence "The numbers of each receptor type in the three ATFL sections were counted twice 77 each by two trained researchers (JYL, YSC)" should not constitute a part of the statistical analysis paragraph, but a part of the procedure previously explained.
: :We made corrections as pointed out by reviewers. thank you
RESULTS
Arrows are difficult to see In figure 2. Please make them bigger and with a different color.
: We have corrected the picture as pointed out by the reviewer. thank you
P-values should be limited to 3 decimals.
: We made corrections as pointed out by reviewers. thank you
DISCUSSION
The results are poorly discussed. Why Is Important to know how many mechanoreceptors there Is In each section of the ligament? Which are the clinical Implications of these findings?
: Thank you for the reviewer's thoughtful comments. Overall, the discussion part was revised by reviewing other literature, and the reason why propioception is necessary and how it can be applied to clinical practice in the future have been added.. It is thought that orthopedic and rehabilitation experts' research is needed. thank you
CONCLUSIONS
Authors suggest that rehabilitation of the ATFL should be based on the injured part and types of mechanoreceptors affected. How can be rehabilitated an Independent part of the ligament or some types of mechanoreceptors only should be part of the discussion section.
: Thank you for the reviewer's thoughtful comments. Based on the discussion, the conclusion was supplemented. thank you
Thank you again for your thoughtful and great review.
The authors would like to thank the reviewer once again for the excellent advice.

Round 2
Reviewer 2 Report
Dear authors,
Thank you for having gone on the modification suggested. I think the manuscript is now clearer.
Sincerely.